# Characteristics of antimicrobial stewardship programmes in hospitals of Uganda

**Isaac Magulu Kimbowa**[1]*, **Moses Ocan**[1], **Jaran Eriksen**[2,3], **Mary Nakafeero**[4], **Celestino Obua**[5], **Cecilia Stålsby Lundborg**[2], **Joan Kalyango**[6,7]

1 Department of Pharmacology and Therapeutics, Makerere University College of Health Sciences, Kampala, Uganda, 2 Department of Global Public Health, Karolinska Institutet, Stockholm, Sweden, 3 Unit of Infectious diseases/Venhälsan, Stockholm South Hospital, Stockholm, Sweden, 4 School of Public Health, Makerere University College of Health Sciences, Kampala, Uganda, 5 Mbarara University of Science and Technology, Mbarara, Uganda, 6 Clinical Epidemiology Unit, Makerere University College of Health Sciences, Kampala, Uganda, 7 Department of Pharmacy, Makerere University College of Health Sciences, Kampala, Uganda

* jakemagulu@gmail.com

**Data Availability Statement:** The data are available in the Open Science Framework repository (DOI 10.17605/OSF.IO/8TNBZ).

## Abstract

While interest in antimicrobial stewardship programmes (ASPs) is growing in most low- and middle-income countries (LMICs), there is a paucity of information on their adoption or implementation in Africa, particularly Uganda. The study assessed the presence and characteristics of ASPs, implemented antimicrobial stewardship (AMS) strategies and the challenges to their implementation in hospitals in Uganda. We conducted a cross-sectional study among heads of infection prevention committees (IPCs) in regional referral hospitals, general hospitals, and private-not-for-profit (PNFP) hospitals from November 2019 to February 2020. An interviewer-administered questionnaire was used to collect data. We analysed data using descriptive statistics. A total of 32 heads of IPCs were enrolled in the study. Of these, eight were from regional referral hospitals, 21 were from general hospitals, and three were from PNFPs. Most heads of IPC were pharmacists (17/32, 53.1%) with a mean age and standard deviation (sd) of 36.1 (±1.1) years. A formal ASP was adopted or implemented in 14 out of the 32 (44%, 95% CI 26–62) studied hospitals. Thirty out of 32 hospitals implemented at least one type of AMS strategy. Sixty-eight percent (22/32) of the hospitals implemented pre-authorisation and approval as their primary AMS core strategy to optimise antibacterial use. The most commonly reported challenges to the implementation of ASP across all 32 hospitals (with or without ASP) were lack of time for the ASP team (29/32, 90.6%) and lack of allocated funding for antimicrobial stewardship team (29/32, 90.6%). In this study, most hospitals in Uganda implemented at least one AMS strategy despite the low implementation of ASPs in hospitals. The ministry of health needs to sensitise and support the establishment of ASP in hospitals across the country.

## Introduction

Antimicrobial stewardship programmes (ASPs) are multidisciplinary programmes that optimise antibacterial use to improve patient outcomes and safety, thus reducing healthcare-

**Funding:** The above grant number provided is the correct one because the study received funding through the Makerere University-Swedish International Development Agency (SIDA) collaboration (Sida PI0010). The funders had no role in study design, data collection and analysis, decision to publish, or preparation of the manuscript.

**Competing interests:** The authors have declared that no competing interests exist.

**Abbreviations:** AMS, Antimicrobial Stewardship; ASP, Antimicrobial Stewardship Programmes; WHO, World Health Organization; GOU, Government of Uganda; MOH, Ministry of Health.

related costs and reducing the development of antimicrobial resistance (AMR) [1–3]. It is critical to optimise currently used antibacterial agents in communities and hospitals to conserve their efficacy whilst reducing the development of antimicrobial resistance [4]. The strategies for optimisation of antibacterial use have been on the national and international agenda, where numerous professional organisations, including clinical, public health, regulatory and accreditation agencies, have advocated implementing antimicrobial stewardship (AMS) interventions to curb the growing burden of antimicrobial resistance (AMR) [2, 5, 6]. AMS is a set of strategies, policies, guidelines, or tools that optimise antibacterial use and reduce AMR development [7, 8].

A decade after coining the term "antimicrobial stewardship," the Infectious Diseases Society of America (IDSA) and the Society for Healthcare Epidemiology of America (SHEA) released the first guidelines in 2007 for establishing ASPs in hospitals [7, 9–11]. The guidelines formed a blueprint for developing ASP structure, leadership, and governance and laid a procedure for selecting AMS core and supplemental strategies for different hospitals [7, 8, 12]. The IDSA guidelines recommended two core primary strategies that are formulary restriction & pre-authorisation and prospective audit/review with intervention & feedback [13]. These two core strategies were used to characterise ASP in a given hospital depending on their implemented strategy [13]. Post prescription audit and review has been reported in multiple studies to impose a low burden on prescribers' prescription autonomy and does not prevent prompt initiation of antibacterial therapy [12]. The implementation of these primary core AMS strategies and their scalability depends on the capacity of the ASPs in hospitals [3, 14]. Furthermore, the recommendations advocated for supplemental AMS strategies to support the optimisation of antibacterial use [10, 12]. The choice of a particular strategy depends on resources and expert availability in the hospitals [8, 15]. Several studies have shown that AMS strategies implemented vary amongst hospital ASPs, with the number and type of personnel involved in the programme influencing the type of strategies, whose choice depended on resources and expert availability in the hospitals [8, 15].

Over the last two decades, hospital ASPs have helped reduce antibacterial consumption and improve the selection of antibacterials [16, 17]. The implementation of the programmes in communities have also increased public awareness of antibacterial use, minimised overuse in animals and human, improved clinical outcomes and reduced the burden of AMR [6, 12]. The emergency of the first guidelines on ASP from IDSA supported the development of regulatory frameworks on ASPs and their implementation in every hospital that needed accreditation in high-income countries than in low-and middle-income countries (LMICs) [7, 18]. However, to support the development of ASPs in LMICs, the World Health Organisation (WHO) developed a Global Action Plan (GAP) on AMR. The GAP on AMR emphasised ASPs implementation in all member countries' human and animal sectors as a key intervention to combat AMR through optimising antibacterial use [19]. Additionally, they also developed a tool kit to guide the adoption or development of ASPs structure, core elements, procedures and strategies at the hospital and country-level among LMICs [20]. Over two-thirds of the member nations of WHO endorsed the GAP on AMR and have implemented national action plans on AMR emphasising ASP implementation in hospitals after the endorsement of the GAP on AMR [21].

However, despite the growing interest in adopting or implementing ASPs in LMICs, there is still slow adoption of ASPs in hospitals [22–27]. Information regarding the availability and characteristics of ASPs in Uganda hospitals remains unknown despite the capacity development of healthcare professionals in implementing AMS interventions and the ongoing implementation of NAP on AMR [28, 29]. Therefore, the study aimed to determine the presence and characteristics of ASP, implemented AMS strategies and the challenges to their

implementation in regional referral hospitals (RRH), general (district) hospitals and tertiary PNFPs in Uganda. The findings will provide the information required to implement ASPs in resource-limited settings.

## Material and methods

### Study setting

We conducted this study at regional referral, general hospitals and tertiary PNFPs for sixteen weeks between November 2019 to February 2020 in Uganda (Fig 1) setting). Over 45% health facilities are public, 40% are PNFPs and 15% private for-profit (PFP) [30, 31]. Public health facilities are hierarchical in terms of referrals and comprise Health Center I-IV, followed by General Hospitals (GH), Regional Referrals hospitals (RRH), and National Referral Hospitals (NRH). They offer free health service delivery. The PNFP offer subsidised service compared to PFP since they receive government funding [30]. Four PNFP hospitals offer specialised services to the urban population, while 40 general hospitals PNFPs serve communities.

In 2020, healthcare professionals from public health facilities, PNFPs and communities were trained to implement AMS strategies to optimise antibacterial use in human and animal sectors [28]. Furthermore, the ministry of health established medicines and therapeutics committees (MTCs) and IPC as systems strengthening exercise to improve medication use in higher public health facilities and PNFPs [29]. To date, there is limited information on characteristics of ASP in regional referral, general hospitals and PNFP. These three types of hospitals (regional referral, general hospitals and PNFPs) were selected because they are expected to develop and implement ASPs and implement AMS strategies after numerous capacity-building initiatives [28].

### Study design and participants

We conducted a cross-sectional study. The study respondents included heads or chairs of IPCs purposively selected because their expertise and experience spanned from administration, research, infection control and prevention to MTC participation. Their position would allow them to respond appropriately to antimicrobial stewardship because they attend different hospitals' committees and administration meetings. After receiving administrative clearance from each participating hospital, the study used various communication strategies (emails, phone calls, and personal visits) to increase IPC heads' participation.

### Sampling procedure

We selected eight regional referral facilities out of 16 using simple random sampling (lottery techniques) in Uganda. Additionally, we selected 32 out of 47 general hospitals using simple random sampling (lottery) as described in our earlier study [32]. The selection of the general hospitals was based on prior selected regional referral hospitals. Lastly, we randomly selected three out of four (75%) tertiary PNFPs graded at a regional referral health facility level.

We purposively selected a chair or head from each IPC of the selected regional referral hospitals, general hospitals and tertiary PNFP.

### Variables

The outcome variables of this study were the proportion of ASPs, AMS strategies and challenges of implementation of programmes in health facilities. In each health facility, the head of the infection prevention committee was asked whether they had a formal ASP programme and they were provided with a response of either "yes" or "no" to the question. However, the study

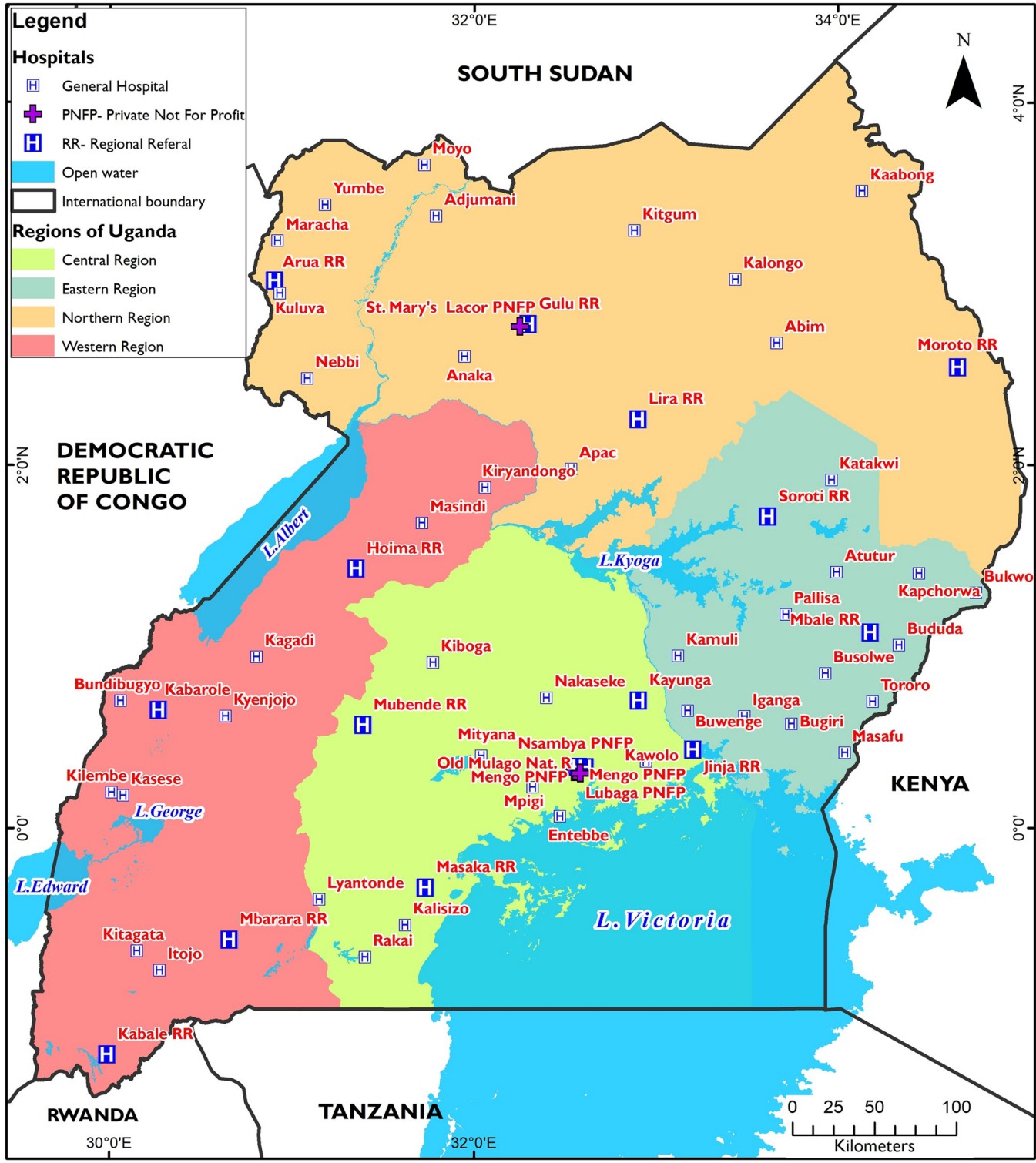

**Fig 1. Map showing hospitals included in the study setting.**

tool never provided a standard definition of an ASP because no formal guidelines or policies require any hospital to develop an ASP. However, hospitals were classified as having an ASP after each of the heads of the infection control committee of the given hospital responded yes on the first cardinal question of whether they had an ASP or not after. AMS strategies were categorized into two (policy items), two (core items), and 14 (supplementary items) which were assessed in hospitals with and without ASP. Core strategies were defined as mandatory strategies adopted or implemented in the ASP, and these are abided by all hospital departments. Policy AMS strategies were defined as strategies adopted from national policy and implemented by the health facility to improve antibiotic use in the hospitals. Supplementary strategies are optional strategies that can be adopted in addition to core strategies to optimise antibacterial use in health facilities. The item statements for AMS strategies were measured using yes coded as 1 or no coded as 0. The challenges of implementing an ASP had 19 item statements to which responses were measured using a scale of "yes" coded as 1 or "no" coded as 0. We assessed the challenges to implementing ASPs in hospitals with existing programmes and those without a programme. Independent variables included hospital characteristics such as the type of health facility (general, regional referral, and private-not for-profit), teaching status (teaching and non-teaching hospital) and bed capacity.

## Questionnaire development

The questionnaire was developed in English using information obtained from previous studies [3, 10, 12, 20]. The questionnaire (S3 File) was developed in consultation with ten experts on infectious diseases (2), epidemiology (1), microbiology (1), pharmacology (3), and public health (3). We used workshops and meetings to enable the ten experts to agree on the sections of the draft questionnaire, item or number of questions, comprehensiveness and clarity of the questions in every section, and item relevance in the questionnaire before the questionnaire was pre-tested. The questionnaire was pre-tested among twenty healthcare providers who were members of the infection control committee to check for clarity and comprehension. Information obtained from the pretesting was used to modify the data collection tool. The tool was used to collect information in the following areas; (i) hospital adopted policy action to contain antibacterial resistance, (ii) antimicrobial stewardship attitudes and practices, (iii) characteristics of antimicrobial stewardship programmes and challenges. Cronbach alpha for AMS strategies was 0.8516, while for challenges of implementing ASP was 0.7716.

## Data collection

We collected data from heads of IPC using an interviewer-administered questionnaire. Research assistants such as pharmacists, medical officers, nurses, and hospital statisticians were trained on the questionnaire before data collection. Potential participants were approached, informed about the study, and requested for their voluntary participation in the study.

The following sections of the questionnaire were used to gather data: hospital characteristics (section A), demographic and general information (section B), ASPs and AMS strategies (section F), and challenges of implementing ASP (section G) (section I). During data collection, the first question asked of participants was whether the hospital had a formal ASP (formally approved and recognised by the hospital administration).

## Data processing and management

The research assistant evaluated every questionnaire for accuracy and completeness at the end of each interview. During data cleaning, we performed a thorough case analysis to detect missing data on variables in the questionnaires. We dropped any questionnaires containing

significant missing data in their variables during the data cleaning process. We utilised Epi-DATA manager version 4.2 to conduct double data entry and validation, during which data collection tools were entered twice by different data entrants, which we reconciled to detect any differences or discrepancies. We performed data validation during data entry to reconcile any files that had deviated from the original files.

## Data analysis

Data were analysed using STATA 15.1 (StataCorp, Texas, USA). We summarised categorical variables using frequencies and proportions, while continuous variables used means and standard deviations. Using chi-square or Fisher exact test, we conducted the univariate analysis to identify the AMS strategies implemented and the challenges associated with adopting ASP between hospitals with ASP and those without ASP. P-values less than 0.05 were considered statistically significant. Estimates of ASP in hospitals were presented using percentages and their corresponding 95% confidence intervals.

## Ethical approval and consent to participate

Makerere University School of Biomedical Sciences Higher Degree Research and Ethics Committee approved the study (reference number SBH-HDREC-624). The Uganda National Council of Science and Technology (UNCST) granted further ethical approval the project (reference number HS339ES). Heads of health facilities granted the study protocol administrative clearance, permitting the principal investigator to conduct the study among healthcare providers in all participating health facilities. Before responding to the questionnaire, written informed consent was obtained from all potential respondents. We kept all the questionnaires collected from the survey in lockable lockers for confidentiality. All information about the healthcare providers was de-identified to ensure anonymity.

## Results

### Characteristics of respondents

A total of 32 out of 43 selected hospitals participated, generating a response rate of 74.4%. Most heads of IPCs who responded were pharmacists (17/32,53.1%), followed by medical specialists (6/32,18.8%), medical officers (5/32, 15.6%), and pharmacy technicians (4/32, 12.5%). The mean age and standard deviation (SD) of IPC heads who responded were 36.1 (±1.1) years. Most heads of IPC in hospitals that participated had a bachelor's degree as their highest level of education. District hospitals had the highest proportion of participation (66%), followed by regional referral (25%) and, lastly, PNFPs (9%) (**Table 1**).

### Characteristics of antimicrobial stewardship programmes in hospitals

ASPs were implemented in 14 of the 32 (44%, 95% CI 26–62) studied hospitals. Five of the fourteen (36%) hospitals with ASP were located in the eastern region, followed by western 29% (4/14), central with 21% (3/14), and northern with 14% (2/14). Half of the hospitals (7/14, 50%) with ASP were regional referral hospitals. Over 63% (9/14) of hospitals with ASP were teaching hospitals. Eight out of the 14 (57%) hospitals with ASP had a bed capacity of 300 beds and above (Table 2).

### Antimicrobial stewardship strategies reported in hospitals

Documentation of antibacterial use in medical charts was the most implemented (30/32, 93.3%) among the two AMS policy strategies (documentation of antibacterials use in medical

**Table 1. Characteristics of the study population (N = 32).**

| Description | Frequency | Percentage |
|---|---|---|
| | (N = 32) | 100% |
| **Type of cadre** | | |
| Pharmacists | 17 | 53.1 |
| Medical officers | 5 | 15.6 |
| Medical specialist | 6 | 18.8 |
| Pharmacy Technicians | 4 | 12.5 |
| **Type of health facility** | | |
| Regional referral hospitals | 8 | 25 |
| District hospitals | 21 | 65.6 |
| Private not for profit | 3 | 9.4 |
| **Teaching status of the hospital** | | |
| Teaching hospitals | 11 | 34.4 |
| Non-teaching hospitals | 21 | 65.6 |
| **Region** | | |
| Central | 10 | 31.3 |
| North | 4 | 12.5 |
| East | 9 | 28.1 |
| West | 9 | 28.1 |
| **hospital bed capacity** | | |
| 100 beds | 21 | 65.6 |
| 101–300 | 3 | 9.4 |
| Over 300 | 8 | 25 |

**Table 2. Characteristics of hospitals with or without an antimicrobial stewardship programme (N = 32).**

| | Hospital with ASP | Hospitals without ASP | Total |
|---|---|---|---|
| | n = 14 | n = 18 | N = 32 |
| | n(%) | n(%) | n(%) |
| **Region** | | | |
| Central | 3 (21.4) | 7 (38.9) | 10 (31.3) |
| North | 2 (14.3) | 2 (11.1) | 4 (12.5) |
| East | 5 (35.7) | 4 (22.2) | 9 (28.1) |
| West | 4 (28.6) | 5 (27.8) | 9 (28.1) |
| **Type of health facility** | | | |
| Regional referral hospital | 7 (50) | 1 (5.6) | 8 (25) |
| District hospital | 5 (35.7) | 16 (88.9) | 21 (65.6) |
| Private-not-for-profit | 2 (14.3) | 1 (5.6) | 3 (9.4) |
| **Nature of the health facility** | | | |
| Teaching hospital | 9(64.3) | 2 (11.1) | 11 (34.4) |
| Non-teaching hospital | 5(35.7) | 16 (88.9) | 21 (65.6) |
| **Bed capacity** | | | |
| 100 beds | 5 (35.7) | 14 (77.8) | 21 (65.6) |
| 101–300 | 1 (7.1) | 2 (11.1) | 3 (9.4) |
| Over 300 | 8 (57.1) | 2 (11.1) | 8 (25) |

Abbreviations, ASP: Antimicrobial Stewardship programmes, %: percent

charts and antibacterial time-out after 48 to 72 hours). All 14 (100%) hospitals with ASP documented antibacterials use in medical charts, as compared to 11 out of 18 (89%) without ASP, though not significantly different (P = 0.198).

Pre-authorisation and approval were the most frequently implemented (22/32, 68.8%) of the two core AMS strategies, with 11 out of 14 (78.6%) hospitals with ASP implementing the AMS strategy, compared to 11 out of 18 (61.1%) hospitals without ASP (P = 0.290).

In all 32 hospitals, antimicrobial combination therapy was the most commonly used (31/32, 96.9%). Antimicrobial combination therapy was implemented in all 14 (100%) hospitals with ASP, compared to 17 out of 18 (94.4%) hospitals without ASP (P = 0.370).

There was a statistically significant difference between hospitals with and without ASP in terms of implementing AMS strategies such as antibacterial time-out (P = 0.014) and supplementary AMS strategies such as streamlining or de-escalation (P = 0.025), reporting of culture and sensitivity results (P = 0.016), and hospital-developed clinical decision-support systems (P = 0.043).

Antibiograms were the least performed supplementary AMS strategy in 9% (3/32) of hospitals. Only two out of 14 (14.3%) hospitals with ASP developed antibiograms, while one out of 18 (5.6%) hospitals without ASP developed antibiograms (**Table 3**).

## Challenges in the implementation of an ASP in selected hospitals in Uganda

In this study, the most reported ASP adoption or implementation challenges included a lack of time 91% (29/32) and dedicated funds for the ASP team 91% (29/32). (**Table 4**).

**Table 3. Antimicrobial stewardship strategies reported in hospitals with or without ASP in Uganda (N = 32).**

| | Hospitals with ASP | Hospitals without ASP | Total | P-value |
|---|---|---|---|---|
| | (n = 14) | n = 18 | N = 32 | |
| | n(%) | n(%) | N (100%) | |
| **1) AMS Policy strategies** | | | | |
| Documentation of antibacterials use in medical charts | 14 (100) | 16 (88.9) | 30 (93.8) | 0.198 |
| Antibacterial time-out after 48 to 72 hour | 10 (71.4) | 5 (27.8) | 15 (46.9) | 0.014* |
| **2) AMS Core strategies** | | | | |
| Pre-authorization and approval technique | 11 (78.6) | 11 (61.1) | 22 (68.8) | 0.290 |
| Prospective audit with Feedback | 9 (64.3) | 8 (44.4) | 17 (53.1) | 0.265 |
| **3) Supplemental AMS strategies** | | | | |
| Antimicrobial order forms | 7 (50) | 4 (22.2) | 11 (34.4) | 0.101 |
| Antimicrobial combination therapy | 14 (100) | 17 (94.4) | 31 (96.9) | 0.370 |
| Streamlining or de-escalation (discontinuing treatment if no bacterial infection | 11 (78.6) | 7 (38.9) | 18 (56.3) | 0.025* |
| Dose optimisation of antibacterial | 13 (92.9) | 12 (66.7) | 25 (78.1) | 0.075 |
| A systematic plan for conversion of parenteral to oral (Intravenous (IV) to oral (PO)) | 13 (92.9) | 14 (77.8) | 27 (84.4) | 0.244 |
| Standard treatment guidelines and clinical pathways | 13 (92.9) | 16(88.9) | 29 (90.6) | 0.702 |
| Use of education strategies to educate prescribers on appropriate prescribing | 13 (92.9) | 12 (66.7) | 25 (78.1) | 0.075 |
| Diagnostic pathways for patients with reported bacterial infection | 9 (64.3) | 12 (66.7) | 21 (65.6) | 0.888 |
| Developing antibiograms | 2 (14.3) | 1 (5.6) | 3 (9.4) | 0.401 |
| Use rapid diagnostic tests | 9 (64.3) | 10 (55.6) | 19 (59.4) | 0.618 |
| Hospital monitor antimicrobial resistance | 7 (50) | 5 (27.8) | 12 (37.5) | 0.198 |
| Reporting of culture and sensitivity results | 9 (64.3) | 4 (22.2) | 13 (40.6) | 0.016* |
| Education on good antimicrobial prescribing practice and resistance | 12 (85.7) | 14 (77.8) | 26 (81.3) | 0.568 |
| Hospital developed Clinical decision-support systems | 8 (57.1) | 4 (22.2) | 12 (37.5 | 0.043* |

AMS: Antimicrobial stewardship, ASP: Antimicrobial stewardship programme, IV: Intravenous, PO: oral

**Table 4. Challenges of implementing ASP in hospitals with or without an ASP (N = 32).**

| | Hospitals with ASP | Hospitals without ASP | Total | P-value |
|---|---|---|---|---|
| | (n = 14) | n = 18 | N = 32 | |
| | n(%) | n(%) | n(%) | |
| Lack of training and education in antimicrobial stewardship | 13 (92.9) | 14 (77.8) | 27 (84.4) | 0.244 |
| Lack of time amongst antimicrobial stewardship programme | 12(85.7) | 17 (94.4) | 29 (90.6) | 0.401 |
| Lack of dedicated funding for an antimicrobial stewardship | 12(85.7) | 17 (94.4) | 29 (90.6) | 0.401 |
| Lack of leadership to promote antimicrobial stewardship at the facility | 10 (71.4) | 11 (61.1) | 21 (65.6) | 0.542 |
| Lack of support from experienced senior clinicians at the facility | 6 (42.9) | 10 (55.6) | 16 (50) | 0.476 |
| Lack of infectious diseases or microbiology services | 7 (50) | 12 (66.7) | 19 (59.4) | 0.341 |
| Lack of pharmacy resources | 5(35.7) | 9 (50) | 14 (43.8) | 0.419 |
| Lack of willingness from healthcare providers to change their prescribing practices | 8 (57.1) | 10 (55.6) | 18 (56.3) | 0.928 |
| Lack of enforcement by facility management/executive | 7 (50) | 13 (72.2) | 20 (62.5) | 0.198 |
| Lack of an electronic medication management system | 9 (64.3) | 12 (66.7) | 21 (65.6) | 0.888 |
| High level of transient or part-time staff | 5 (35.7) | 8(44.4) | 13 (40.6) | 0.618 |
| Inadequate time for Antimicrobial stewardship activities | 11 (78.6) | 13 (72.2) | 24 (75) | 0.681 |
| Personnel and expertise shortages | 7 (50) | 14 (77.8) | 21 (65.6) | 0.101 |
| Inadequate funding for AMS activities | 11(78.6) | 13 (72.2) | 24 (75) | 0.681 |
| Lower priority for AMS than other clinical initiatives | 11 (78.6) | 15 (83.3) | 26 (81.3) | 0.732 |
| Inadequate information technology (IT) support for antimicrobial stewardship | 11 (78.6) | 13 (72.2) | 24 (75) | 0.681 |
| Opposition of AMS by healthcare providers | 8 (57.1) | 8 (44.4) | 16 (50) | 0.476 |
| A paucity of data on improved outcomes with ASPs in Uganda | 11 (78.6) | 15 (83.3) | 26 (81.3) | 0.732 |
| Lack of policy support for ASP activities from government or other partners | 12 (85.7) | 15 (83.3) | 27 (84.4) | 0.854 |

Abbreviations, ASP: Antimicrobial Stewardship programmes, %: percent

## Discussion

The Uganda national action plan (NAP) on AMR (2018–2023) is under implementation, emphasising optimising antibacterial use through strengthening and operationalising ASP among all hospitals regardless of their size [33]. While many healthcare providers from public and non-profit hospitals have received AMS training to help them operationalise and strengthen ASPs in their facilities, there is limited information on the characteristics of ASP, the AMS strategies used, and the challenges associated with ASP implementation [28].

In this survey, only 4 out of 10 hospitals implemented ASPs in their hospitals. Our study found a higher prevalence of adopted ASPs than a recent study in Nigeria, which found that 35% of the 17 hospitals in six different Nigerian regions implemented ASPs [34]. The difference in the proportion of ASP in our study can be explained by the number of participating hospitals (32 versus 17) and their difference in types and teaching status [34]. Furthermore, our findings show that ASPs were distributed in tertiary hospitals (regional referral hospitals) and district hospitals and PNFPs, contrary to the Nigerian study that only reported ASP in Tertiary hospitals. Previous studies indicated that larger hospitals (tertiary or regional referrals) were more likely to implement ASP than smaller hospitals due to the availability of more resources in the former than the latter, which contrasts our findings [1, 35]. The distribution of ASP in different types of hospitals (regional referrals, district hospitals, and PNFPs) of Uganda could be due to the Ministry of Health and other development partners continued strengthening and operationalisation of ASP and medicines and therapeutics committees in public hospitals and PNFPs. The low adoption of ASPs without any formal guidelines indicates that hospitals have recognised their beneficial effects in optimising antibacterial use, and some

of the trained healthcare providers are spearheading their introduction from the available international guidelines of CDC and WHO and local capacity development from one health initiative [12, 20, 28]. This finding will go a long way in supporting the dissemination strategy of the national action plan on AMR and supporting national guidelines on developing ASP [33].

Regardless of the absence or presence of an ASP, hospitals in our study implemented AMS strategies broadly. At least one AMS strategy was conducted in over 96% of hospitals surveyed. This finding was consistent with previous studies conducted in the United States that reported that over 90% of hospitals implemented at least one AMS strategy despite having an ASP or not [11, 36, 37]. However, it contrasts with previous Nigerian studies, where only 72% of hospitals performed at least one AMS strategy [38, 39]. The majority of the hospitals implemented documenting antibacterial in the medical charts, pre-authorisation and approval of antibacterial and antimicrobial combination therapy as dominant policy, core and supplementary AMS strategies respectively, where the difference in implementing these AMS strategies between hospitals with or without ASP was not statistically significant. Several studies have reported the difficulty in predicting which AMS strategy is effective for the case of those hospitals implementing several strategies though this may have more impact on optimising antibacterial use than the total amount of resources available [36].

In our study, pre-authorisation and approval were the most frequently reported AMS core strategy among the 32 hospitals. This finding is similar to a previous study in Nigeria, where 12 out of 17 tertiary hospitals conducted pre-authorisation as a common core AMS strategy [34]. The drivers influencing hospitals with or without ASP to adopt pre-authorisation as a core AMS strategy in our study may not be clear, but other studies in Africa have alluded to limited resources, thus using the strategy to reserve certain antibacterials among all healthcare providers [34]. However, our findings may contrast the Infectious Diseases Society of America (IDSA) recommendations and findings of previous studies that reported post prospective audit and feedback as the most performed AMS strategy [7, 40]. The strategy was reported to be less cumbersome to prescribers, and it did not create a barrier to initiation of therapy and was more acceptable to prescribers [7, 40]. This finding might imply that policymakers' adoption of core AMS strategy has to consider the availability of a certain staffing level, resources, and region before recommending specific AMS strategies to hospitals.

A lack of dedicated funding and time for the ASP team were the most frequently reported challenges to adopting an ASP in hospitals without ASP and those implementing ASP. A report by the WHO on the ability of LMICs to implement the AMS tool kit highlighted similar issues as a barrier to AMS implementation including a lack of human and financial resources and insufficient training [41]. In addition, other reviews have identified a lack of diagnostic facilities, poor laboratory service infrastructure, a high incidence of infectious illnesses, and a lack of suitable staffing levels as significant barriers to ASP implementation in Uganda [41, 42]. Given this finding, the national steering committee on AMS must consider these issues while developing guidelines and frameworks for ASP implementation in Uganda.

## Limitations

Our study was cross-sectional, with respondents from various types of hospitals in various regions. We used only the head or chair of IPC, thus we could not verify the accuracy or completeness of responses. However, the heads of IPC were privileged because their position enabled them to attend several meetings and be informed on several hospital administrative issues including AMS. This may have controlled for their overestimation or underestimation of responses about the ASP. Even after differentiating between hospitals with and without ASP

coverage, there was no statistically significant difference in adopting core AMS strategies. Our statistical study was descriptive in determining the characteristics of ASP and corresponding AMS strategies in those hospitals with or without ASP. Due to the small sample size, we could not identify factors associated with ASP adoption in Uganda. Despite these limitations, the present study offers insight into hospitals' progress in adopting ASP and AMS strategies regardless of the availability of guidelines on ASP. It further confirms that there has been some benefit in the ongoing capacity building programmes to operationalise ASP in Uganda. The findings highlight the need for the Ugandan government to support ASP in all Ugandan hospitals.

## Conclusions

The majority of hospitals in Uganda have implemented at least one AMS strategy to optimise antibacterial use despite the low coverage of ASP. Pre-authorization was the most adopted AMS core strategy in hospitals despite their ASP status. The ministry of health needs to sensitise and support the establishment of ASP in hospitals across the countries.

## Supporting information

**S1 File. STROBE statement checklist of items included in cross-sectional studies.**
(DOCX)

**S2 File. Informed consent form.**
(DOCX)

**S3 File. Questionnaire.**
(DOCX)

## Acknowledgments

We thank the 32 heads of the infection control committee from the eight regional referrals hospitals, 21 general hospitals, and three private-not for-profit facilities. Finally, we are grateful to all the research assistants who supported us in the data collection in this study. The authors appreciate the valuable input of Prof Jasper Ogwal Okeng and Dr Jackson Mukonzo in this work.

## Author Contributions

**Conceptualization:** Isaac Magulu Kimbowa, Moses Ocan, Jaran Eriksen, Mary Nakafeero, Celestino Obua, Cecilia Stålsby Lundborg, Joan Kalyango.

**Data curation:** Isaac Magulu Kimbowa, Moses Ocan, Mary Nakafeero, Joan Kalyango.

**Formal analysis:** Isaac Magulu Kimbowa, Moses Ocan, Mary Nakafeero, Joan Kalyango.

**Funding acquisition:** Jaran Eriksen, Celestino Obua, Cecilia Stålsby Lundborg.

**Investigation:** Isaac Magulu Kimbowa, Joan Kalyango.

**Methodology:** Isaac Magulu Kimbowa, Moses Ocan, Mary Nakafeero, Celestino Obua, Joan Kalyango.

**Project administration:** Moses Ocan, Jaran Eriksen, Celestino Obua, Cecilia Stålsby Lundborg, Joan Kalyango.

**Resources:** Jaran Eriksen, Celestino Obua, Cecilia Stålsby Lundborg.

**Software:** Mary Nakafeero.

**Supervision:** Moses Ocan, Jaran Eriksen, Celestino Obua, Cecilia Stålsby Lundborg, Joan Kalyango.

**Validation:** Isaac Magulu Kimbowa, Moses Ocan, Mary Nakafeero, Cecilia Stålsby Lundborg, Joan Kalyango.

**Visualization:** Isaac Magulu Kimbowa, Moses Ocan.

**Writing – original draft:** Isaac Magulu Kimbowa.

**Writing – review & editing:** Moses Ocan, Jaran Eriksen, Celestino Obua, Cecilia Stålsby Lundborg, Joan Kalyango.

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
