## [Decision Letter · Decision Letter 0]

7 Feb 2022

PONE-D-21-34980CHARACTERISTICS OF ANTIMICROBIAL STEWARDSHIP PROGRAMMES IN HOSPITALS OF UGANDAPLOS ONE

Dear Dr. Kimbowa,

Thank you for submitting your manuscript to PLOS ONE. After careful consideration, we feel that it has merit but does not fully meet PLOS ONE’s publication criteria as it currently stands. Therefore, we invite you to submit a revised version of the manuscript that addresses the points raised during the review process.

We look forward to receiving your revised manuscript.

Kind regards,

Ismaeel Yunusa, PharmD, PhD

Academic Editor

PLOS ONE

Journal Requirements:

“No-The funders had no role in study design, data collection and analysis, decision to publish, or preparation of the manuscript.”

“This study obtained funding through the Makerere University-Swedish International Development Agency (SIDA) collaboration (Sida PI0010). However, the funders never participated in the study design, data collection and analysis, publishing decisions, or manuscript preparation.”

“No-The funders had no role in study design, data collection and analysis, decision to publish, or preparation of the manuscript.”

Reviewers' comments:

Reviewer's Responses to Questions

**Comments to the Author**

1. Is the manuscript technically sound, and do the data support the conclusions?

Reviewer #1: Partly

Reviewer #2: Yes

2. Has the statistical analysis been performed appropriately and rigorously? 

Reviewer #1: Yes

Reviewer #2: Yes

3. Have the authors made all data underlying the findings in their manuscript fully available?

Reviewer #1: Yes

Reviewer #2: Yes

4. Is the manuscript presented in an intelligible fashion and written in standard English?

Reviewer #1: Yes

Reviewer #2: No

5. Review Comments to the Author

Reviewer #1: Thank you for the opportunity to review your manuscript. Please see below a few comments to further enhance your manuscript.

1.Abstract and introduction

•Please ensure all acronyms and abbreviations are defined at first mentioned. You also need to be consistent in the use of these acronyms/abbreviations.

2.Methodology

•There is a discrepancy in the survey timeline. In the abstract you stated that the survey was conducted b/w Oct to Feb 2020 but in the methodology under study setting you’ve written that this was b/w Nov to Feb 2020. Please clarify. You’ve also indicated that the survey was conducted over six weeks, yet the two time periods you have written down are far longer than this. Please clarify.

•The sampling process you have described seems to be more along the lines of a purposive sample comprising three cohorts as opposed to the three staged sampling you have stated in your manuscript. The fact that you have conditionally selected hospitals based on proximity and referral shows that your sample is unlikely to have been truly random. Further, it would have been impractical to randomly select three hospitals out of four as indicated in the third cohort in your sample. Overall, you will need to think hard on the sampling process and describe this appropriately.

•Your thought appears to have trailed off in lines 122- 124 as the sentence is incomplete. Please review

•In lines 130 – 132, you mentioned you did not provide a definition of what constitutes an ASP. I therefore wonder how ASP was operationalised in the study to ensure that your respondents clearly understood what this was.

•Lines 149 – 152, please state the number of experts and workshops involved in the development of your questionnaire.

•Lines 176 – 177, how many questionnaires were discarded using the process you have described? Please provide full details.

3. Results

No comments

4. discussion

please do not quote and report your study results in this section as this will be repetitive

5. Conclusions

No comments

Best wishes.

Reviewer #2: There are a number of ambiguities and grammatical errors in the manuscript which need to be addressed. I have added comments and highlighted the relevant sections in the manuscripts for the authors' attention. The manuscript with enclosed comments will be uploaded.

6. PLOS authors have the option to publish the peer review history of their article (what does this mean?). If published, this will include your full peer review and any attached files.

Reviewer #1: No

Reviewer #2: No

---

## [Author Response · Author response to Decision Letter 0]

11 Apr 2022

Dear Editor: 

We appreciate the comments. We have revised our manuscript according to the editor and reviewers' comments, questions, and suggestions. We appreciate the comments and feedback since they have improved our manuscript. We are grateful for your service. 

Yours, 

Isaac Kimbowa 

Corresponding author

---

## [Editor Report · Decision Letter 1]

21 Apr 2022

CHARACTERISTICS OF ANTIMICROBIAL STEWARDSHIP PROGRAMMES IN HOSPITALS OF UGANDA

PONE-D-21-34980R1

Dear Dr. Kimbowa,

We’re pleased to inform you that your manuscript has been judged scientifically suitable for publication and will be formally accepted for publication once it meets all outstanding technical requirements.

Kind regards,

Ismaeel Yunusa, PharmD, PhD

Academic Editor

PLOS ONE

Additional Editor Comments (optional):

May need to correct some table formatting
---

## [Editor Report · Acceptance letter]

29 Apr 2022

PONE-D-21-34980R1 

CHARACTERISTICS OF ANTIMICROBIAL STEWARDSHIP PROGRAMMES IN HOSPITALS OF UGANDA 

Dear Dr. Kimbowa:

I'm pleased to inform you that your manuscript has been deemed suitable for publication in PLOS ONE. Congratulations! Your manuscript is now with our production department. 

Kind regards, 

on behalf of

Dr. Ismaeel Yunusa 

Academic Editor

PLOS ONE